# Solid Lipid Nanoparticles as Carriers for the Synthetic Opioid LP2: Characterization and In Vitro Release

Angelo Spadaro [1,*], Lorella Pasquinucci [1], Miriam Lorenti [1], Ludovica Maria Santagati [2], Maria Grazia Sarpietro [2], Rita Turnaturi [1], Carmela Parenti [3] and Lucia Montenegro [2,*]

[1] Department of Drug and Health Sciences, Medicinal Chemistry Section, University of Catania, Viale Andrea Doria 6, 95125 Catania, Italy; lpasquin@unict.it (L.P.); miriam.lorenti@gmail.com (M.L.); rita.turnaturi@unict.it (R.T.)

[2] Department of Drug and Health Sciences, Pharmaceutical Technology Section, University of Catania, Viale Andrea Doria 6, 95125 Catania, Italy; ludovica.santagati@icloud.com (L.M.S.); mg.sarpietro@unict.it (M.G.S.)

[3] Department of Drug and Health Sciences, Pharmacology Section, University of Catania, Viale Andrea Doria 6, 95125 Catania, Italy; cparenti@unict.it

[*] Correspondence: angelo.spadaro@unict.it (A.S.); lmontene@unict.it (L.M.); Tel.: +39-095-738-4002 (A.S.); +39-095-738-4010 (L.M.)





**Featured Application: The incorporation of the synthetic anti-nociceptive ligand LP2 into solid lipid nanoparticles would allow its parenteral administration, thus improving its efficacy. As LP2 is a diastereoisomeric mixture of 2R/2S-LP2, the HPLC method, developed and validated in this work to separate and quantify each diastereoisomer, could be useful in further pharmacological studies.**

**Abstract:** A synthetic dual-target mu opioid peptide receptor/delta opioid peptide receptor anti-nociceptive ligand, named LP2, has emerged as a promising candidate for the management of acute and/or persistent pain, but its lipophilicity limits further developments as a therapeutic agent. In this work, to allow designing aqueous formulations of LP2 for parenteral administration, solid lipid nanoparticles (SLNs) were investigated as LP2 nanocarriers. LP2-loaded SLNs were prepared by the phase-inversion temperature method, showing good technological properties (small mean particle, size, low polydispersity index, good stability). As LP2 was a diastereoisomeric mixture of 2R/2S-LP2, an HPLC method was developed to identify and quantify each diastereoisomer, and this method was used to assess LP2 in vitro release from SLNs. The developed method, based on reverse-phase chromatography using an isocratic mobile phase consisting of 50% methanol and 50% triethanolamine at 0.3% (pH = 3 with trifluoroacetic acid), allowed efficient separation of 2R- and 2S-LP2 peaks and reliable quantification with intra- and inter-day precision and accuracy within the acceptability limit, expressed as relative standard deviation set at ≤15%. The results of this study suggest that the incorporation of LP2 into SLNs could be a promising strategy to design suitable formulations for further pharmacological studies involving LP2.

**Keywords:** solid lipid nanoparticles; diastereoisomers; LP2; multitarget ligand; opioids; anti-nociceptive agents; HPLC

## 1. Introduction

In recent decades, solid lipid nanoparticles (SLNs) have been widely investigated as carriers for drug delivery by different administration routes [1–5]. The great deal of interest focused on SLNs has been due to their many advantages compared to other colloidal systems, including increased drug bioavailability and stability, adjustable technological properties, ability to control and/or to target drug delivery, low production cost, and ease scale-up [6,7]. In addition, as SLNs are made up of a solid lipid core stabilized by surfactants in aqueous media, lipophilic drugs could be entrapped in the SLNs matrix,

resulting in an improvement of drug water solubility. Therefore, such nanocarriers provide a useful tool to design aqueous formulations for poorly soluble drugs [8–10].

Recently, a dual-target mu opioid peptide receptor (MOPr)/delta opioid receptor (DOPr) anti-nociceptive ligand, named LP2 [11], containing a benzomorphan nucleus (Figure 1), has been synthesized and investigated for its in vivo biological activities in animal models of acute and persistent pain [12]. The development of multitarget ligands has emerged as a promising strategy to enhance therapeutic efficacy while reducing adverse effects [13–15]. In particular, compounds with multitarget opioid activity are effective anti-nociceptive agents for pain management with a limited incidence of adverse effects, usually associated with clinically used opioid analgesics acting at a single target [16–18]. The improved pharmacological fingerprint of dual-target MOPr/DOPr agonists, which are able to simultaneously target MOPr and DOPr, is related to the co-expression of MOPr and DOPr in key sites for pain modulation, allowing their inter-modulatory interactions, both physically and functionally. LP2 was able to bind to and simultaneously activate MOPr (Ki = 1.08 nM, $IC_{50}$ = 21.5 nM) and DOPr (Ki = 6.6 nM, $IC_{50}$ = 4.4 nM) [11]. In the tail-flick test, LP2 produced a significant long-lasting anti-nociceptive effect, naloxone-reversed, with an $ED_{50}$ of 0.9 mg/kg i.p. in mice. Tests in models of persistent pain showed that LP2 elicited a significant anti-inflammatory effect ($ED_{50}$ = 0.88 and 0.79 mg/kg i.p) in both phases I and II of the formalin test in mice, without motor impairments in the Rotarod test [19]. In addition, in unilateral sciatic nerve chronic constriction injury (CCI) neuropathy in rats, LP2 ameliorated mechanical allodynia signs from the early phase of treatment up to 21 days post-ligatures [20].

R = 2*R/S*-OCH₃

**Figure 1.** Chemical structure of (2*R/S*)-N-2-methoxy-2-phenylethyl-6,7-benzomorphan (LP2).

Unfortunately, the lipophilic nature of LP2 is a major concern for its further development as a new therapeutic anti-nociceptive agent. As reported in the literature [21], many (more than 40%) drug candidates are poorly soluble in water [21]; therefore, several strategies have been investigated to overcome this drawback, including chemical modifications of drugs and/or incorporation into suitable delivery systems [8–10].

To address the issue of LP2's low water solubility allowing the design of formulations for parenteral administration, in the present work, this dual-target MOPr/DOPr anti-nociceptive ligand was incorporated into SLNs, and the resulting colloidal dispersions were assessed to determine their technological properties (mean particle sizes, polydispersity index, zeta potential, morphology, and stability) and in vitro LP2 release.

As LP2 is a diastereoisomeric mixture due to the presence of a stereocenter at the (R/S)-2-methoxy-2-phenylethyl group as an N-substituent of the 6,7-benzomorphan scaffold, a further aim of this work was the development of an effective analytical method to identify and quantify each LP2 diastereoisomer.

The availability of a suitable and reliable analytical method was fundamental to assess the release rate of each diastereoisomer after loading LP2 as a racemic mixture into SLNs.

In addition, it is well known that, despite their identical molecular formulas, atom-to-atom linkages, and bonding distances, diastereoisomers could differ in their pharmacokinetic and pharmacodynamic profiles because, in living beings, drug targets could be chiral, as well.

Therefore, the development of an effective analytical method to determine LP2 diastereoisomers could be a valuable tool in further pharmacological studies on this dual-target MOPr/DOPr agonist.

## 2. Materials and Methods

### 2.1. Materials

Water, acetonitrile, and methanol (HPLC grade) were from Merck (Milan, Italy). LP2, 2R- and 2S-LP2 diastereoisomers standard were kind gifts from Prof. Lorella Paquinucci and were synthesized as previously reported [11,22].

Cetyl palmitate (CP) and glyceryl oleate (GO) were bought from ACEF (Fiorenzuola D'Arda, Italy). Polyoxyethylene-20-oleyl ether (oleth-20) was purchased from Sigma-Aldrich srl (Milan, Italy). All other chemicals were of reagent grade.

### 2.2. Preparation of Solid Lipid Nanoparticles (SLNs)

SLNs were prepared using the phase-inversion temperature (PIT) as previously reported [23,24]. The oil phase components were CP (7,0% w/w, solid lipid), oleth-20, (8.7% *w/w*, surfactant), GO (4.4% *w/w*, co-surfactant), and LP2 (1.1% *w/w*), while the aqueous phase consisted of saline solution (NaCl 0.90% *w/w*). Unloaded SLNs (without the addition of LP2, SLN A) were prepared as control. The oil phase and the aqueous phase were separately heated, and when both phases were at about 90 °C, the aqueous phase was slowly added to the oil phase under continuous stirring (700 rpm). The resulting mixture was allowed to cool to room temperature under mixing, and the PIT value (temperature at which the turbid mixture turned clear) was recorded using a conductivity meter (model 525, Crison, Modena, Italy). The colloidal suspensions were filtered with syringe filters (cellulose acetate, 0.20 μm, sterile, LLG, Meckenheim, Germany) and stored in airtight jars at room temperature and sheltered from light until used.

### 2.3. Dynamic Light Scattering (DLS)

The mean particle size and the size distribution (polydispersity index, PDI) were determined by DLS using a Zetasizer Nano ZS90 (Malvern Instruments, Malvern, UK), equipped with a laser diode (4 mW, 670 nm) scattering light at 90°.

Samples (diluted 1:5 using distilled water) were thermostated at 25 °C for 2 min prior to the analysis. The same instrument was used to determine $\zeta$-potential by laser Doppler velocimetry after sample dilution using KCl 1 mM (pH 7.0) prior to the analysis. All measurements were carried out in triplicate, and the results were expressed as mean $\pm$ SD.

### 2.4. Transmission Electron Microscopy (TEM)

TEM analyses were performed using a transmission electron microscope (model JEM 2010, Jeol, Peabody, MA, USA) operating at an acceleration voltage of 200 KV. Colloidal suspensions (5 μL) were placed on a Formvar (200-mesh) copper grid (TAAB Laboratories Equipment, Berks, UK). When the sample was absorbed, the surplus was removed by filter paper, and an aqueous solution of uranyl acetate (2% *w/w*) was added. The sample was allowed to dry at 25 °C, and TEM images were acquired.

### 2.5. Differential Scanning Calorimetry (DSC) Analyses

Calorimetric analyses were performed by a Mettler-Toledo STARe system (Mettler-Toledo, Greifensee, Switzerland) equipped with a DSC-822e calorimetric cell and a Mettler TA-STARe software to analyze the obtained data. The sensitivity was automatically chosen as the maximum allowed by the calorimetric system. The reference pan was filled with the same solvent as the samples under study. The calorimetric system was calibrated,

in temperature and enthalpy changes, following the procedure of the DSC 822 Mettler TA STARe instrument. A total of 100 μL of sample was put into the calorimetric pan, hermetically sealed, and submitted to analysis as follows: (i) a heating scan from 5 to 85 °C (2 °C/min); (ii) a cooling scan from 85 to 5 °C (4 °C/min), at least three times. Each analysis was carried out in triplicate. The melting enthalpy variation was obtained by integration of the area under the transition peak.

### 2.6. Stability Tests

Particle sizes, PDI, and $\zeta$-potential values of SLNs samples were measured at intervals (24 h, one week, two weeks, one month, two months). During storage, samples were maintained at room temperature and sheltered from light exposure.

### 2.7. In Vitro Release Experiments

LP2 release from SLNs was determined using dialysis membranes (Spectra/Por™ CE, diameter 29 mm, Mol. Wt. cutoff: 3000, Spectrum, Los Angeles, CA, USA), previously moistened by immersion in water for 24 h at room temperature. The dialysis medium consisted of water/ethanol (80/20 $v/v$) to increase LP2 solubility in the receptor medium, thus ensuring pseudo-sink conditions during the experiments. The same receiving phase had already been used in previous studies to evaluate in vitro release from SLNs of lipophilic drugs and did not lead to any alteration of nanoparticle structural properties [15]. The dialysis medium was stirred (700 rpm) and thermostated at 37 °C during the experiment. A total of 1000 μL of each formulation was pipetted into a dialysis membrane bag that was sealed and immersed in a beaker containing 65 mL of the release medium. Unloaded SLNs were used as control. At intervals, samples (200 μL) of the release medium were withdrawn and replaced with an equal volume of receiving solution pre-equilibrated at 37 °C. Each experiment was run for 24 h and sheltered from light to prevent any photo-degradation. Each experiment was carried out in duplicate. The withdrawn samples were analyzed to determine LP2 content by the HPLC method developed in this study.

### 2.8. High-Performance Liquid Chromatography (HPLC) Analyses

HPLC analyses were performed on an Agilent 1260 Infinity II chromatographic system (Agilent Technologies, Cernusco sul Naviglio, Italy) equipped with HPLC ChemStation OpenLab software (M8307AA), a quaternary pump G7111B, a diode array detector (DAD) G7115A, a manual sample injector (G1328C) with a 20 μL loop, and a thermostated column compartment G1316A.

The HPLC method was developed on a Knauer Eurosphere II 100-3 C18 (150 mm × 4.60 mm, 5.0 μm) using isocratic binary mobile phases consisting of 50% methanol and 82% triethylamine 0.3% in water (pH adjusted to 3.0 with trifluoroacetic acid).

The flow rate was 1.0 mL/min, the column temperature was 22 °C, and the detection wavelength was 280 nm. Each analysis was run in triplicate. UV spectra were recorded in the range 200–400 nm, and chromatograms were acquired at 280 and 254 nm.

#### 2.8.1. Determination of LP2 Content in SLNs

A total of 100 μL of the colloidal suspension was diluted 1:10 with methanol and sonicated for 30 min. Subsequently, 100 μL of the obtained mixture was diluted 1:10 with the mobile phase, filtered (0.45 μm Spartan filters, Schleicher and Schuell, Dassel, FRG), and injected into the HPLC system. The possible absorption of the filter was excluded comparing, by HPLC analysis, the same sample subjected to a double treatment: centrifuged plus filtered vs. centrifuged only.

#### 2.8.2. Calibration

A stock solution of 2*R*-LP2 and 2*S*-LP2 (150.0 μg/mL) was obtained by dissolving an appropriate amount of the single standard in mobile phase. Working standard solutions

of 2R-LP2 and 2S-LP2 were prepared daily by adequate dilution with the eluent phase of calculated amount of the stock solution.

Seven-point calibration curves were set up for both 2R- and 2S-LP2 standards to test the linearity of the UV-DAD response. Calibration standards were processed as reported in the above-mentioned sample-preparation procedure and analyzed by HPLC.

### 2.8.3. Validation

The developed HPLC method was validated according to International Conference on Harmonization Guidelines [25] with regard to linearity, limit of detection (LOD) and quantitation (LOQ), precision (intra-day and inter-day), and accuracy, as previously reported [26].

### *2.9. Determination of the Lipophilic Index*

The lipophilic index (log K) of each LP2 diastereoisomer was determined by the HPLC method described above using the following equation:

$$\log K = \log (t_r - t_0)/t_0 \tag{1}$$

where $t_r$ is the retention time of the retained peak and $t_0$ is the retention time of an elution solvent.

### 3. Results and Discussion

Unloaded SLNs, prepared by the PIT method, showed a small mean particle size, which was not affected by LP2 loading (see Table 1). PDI values were lower than 0.300, suggesting the presence of a homogeneous population of nanoparticles in the investigated samples. Analogously to mean particle size and PDI, ζ-potential values were similar for unloaded and LP2-loaded SLNs. As reported in the literature [27], ζ-potential values greater (as an absolute value) than 30 mV are needed to provide stable colloidal dispersions. However, unloaded and LP2-loaded SLNs showed good stability during storage for two months at room temperature, as no significant change of particles size, PDI, and ζ-potential values was observed (data not shown). All samples remained clear during the storage period with no sign of drug precipitation. SLNs with similar technological properties (small mean size, low PDI and ζ-potential values, and good stability during storage at room temperature) were obtained in previous works using the PIT method [23]. The presence of long polyoxyethylene chains of the surfactant oleth-20 on the nanoparticle surface, which could provide a steric stabilization, was considered responsible for the good stability of the resulting nanocarriers [23].

**Table 1.** Mean size (size), polydispersity index (PDI), phase inversion temperature (PIT), and ζ-potential (Zeta) of unloaded (SLN A) and LP2-loaded SLNs.

| Sample | Size ± S.D. (nm) | PDI ± S.D. | PIT (°C) | Zeta (mV) |
|---|---|---|---|---|
| SLN A | 27.22 ± 2.06 | 0.128 ± 0.030 | 65 | −10.5 |
| SLN LP2 | 29.79 ± 1.50 | 0.126 ± 0.029 | 71 | −10.7 |

As illustrated in Table 1, LP2-loaded SLNs showed a PIT value greater than that of unloaded SLN, thus suggesting different interactions among the lipid core components in the presence of LP2. Therefore, DSC studies were carried out to assess the thermal behavior of the investigated SLNs. The LP2 calorimetric peak was centered at 57.78 °C. The calorimetric curve of unloaded SLN was characterized by the main peak at 41.90 °C, and a shoulder at a lower temperature was present (Figure 2). Such shoulder could indicate that the surfactant was not homogeneously distributed in the SLNs' structure.

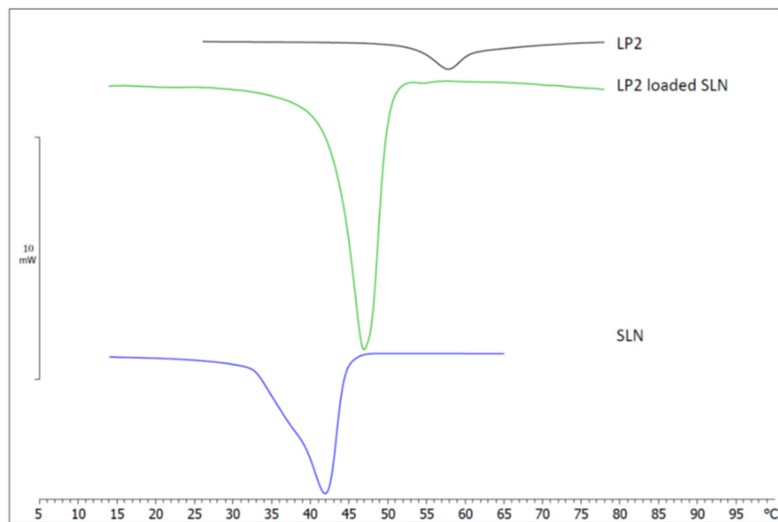

**Figure 2.** Calorimetric curves of LP2, unloaded solid lipid nanoparticles (SLNs), and LP2-loaded solid lipid nanoparticles (LP2-loaded SLNs).

The calorimetric curve of LP2-loaded SLNs was quite different, as there was no evidence of the previously mentioned shoulder, and the main peak was sharper and centered at 46.89 °C. These results suggest that the incorporation of LP2 affected the calorimetric behavior of SLNs; in particular, the absence of the shoulder could indicate a homogeneous distribution of the components of the SLN.

Many parameters can be taken into account to study the calorimetric behavior of SLNs. The most important and used are the transition temperature (T), which marks the transition from an ordered state to a disordered state; the enthalpy variation (ΔH); and the ΔT1/2, which can be taken as a measure of the cooperativity of the transition (the number of lipid molecules undergoing simultaneous transition); in particular, ΔT1/2 is inversely proportional to cooperativity. The comparison of these parameters for unloaded and LP2-loaded SLNs could provide information on the effect of LP2 incorporation into SLNs. LP2-loaded SLNs showed a transition temperature increase of about 5 °C; therefore, the presence of LP2 in the SLNs structure favored the ordered state. ΔH was −11.90 J/g in unloaded SLNs and −18.02 J/g in LP2-loaded SLNs and a decrease in ΔT1/2 (calculated by the DSC software as the temperature width at half peak height) from 5.99 °C in unloaded SLN to 4.40 °C in LP2-loaded SLN was observed. These values suggest that LP2 caused an increase in lipid cooperativity. In addition, in LP2-loaded SLNs' calorimetric curve, there was no evidence of the peak at 57.78 °C, characteristic of LP2, thus indicating that the compound lost the crystalline form and was present in an amorphous state in the SLNs' structure.

From a morphological point of view, TEM analyses showed that both unloaded and LP2-loaded SLNs were roughly spherical, with no evident sign of aggregation (Figure 3).

Prior to performing in vitro release experiments, the LP2 content (as a mixture of 2*R* and 2*S* isomers) of SLNs samples was determined by the high-performance liquid chromatography (HPLC) method developed in this study. More than 99% (99.9 ± 0.3%) of the LP2-loaded dose was recovered from SLNs samples, and no significant difference in the LP2 isomers levels was detected. Being LP2-lipophilic, all of the loaded drug was supposed to be incorporated into the lipid core of SLNs; therefore, the loading capacity of such SLNs was considered equal to 1.1% *w/w*. As reported in the literature [28], when poorly water-soluble compounds are incorporated into lipid nanoparticles, if the resulting colloidal dispersion is clear, all of the drug must be located in the lipid phase of the SLN dispersion. Therefore, in a clear colloidal dispersion, drug incorporation could be regarded as approximately 100%.

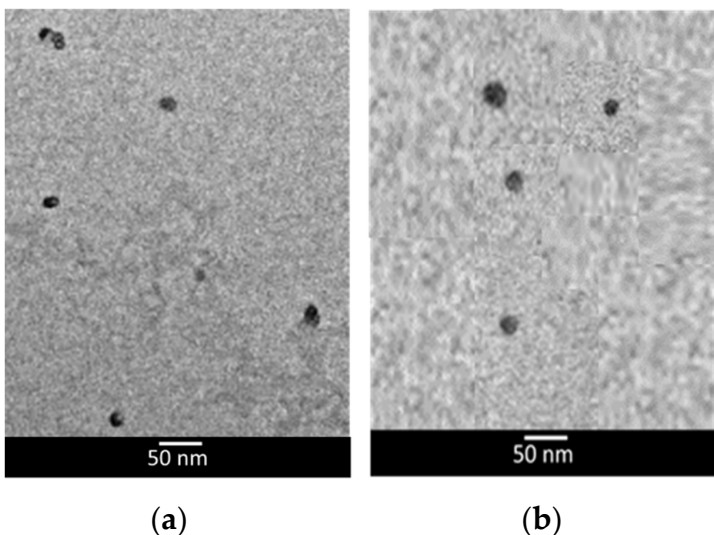

(a)                                    (b)

**Figure 3.** Transmission electron microscopy (TEM) images of unloaded lipid nanoparticles (**a**) and LP2-loaded lipid nanoparticles (**b**).

In vitro release profiles of both LP2 diastereoisomers from the SLNs under investigation are illustrated in Figure 4. LP2 was loaded into SLNs as a racemic mixture because previous studies highlighted that LP2, 2*R*-LP2, and 2*S*-LP2 exhibited a different profile in modulating acute thermal nociception. In particular, the anti-nociceptive effect of LP2 maintained higher threshold values for longer periods compared to both isomers, a profile especially useful in persistent pain treatment [11].

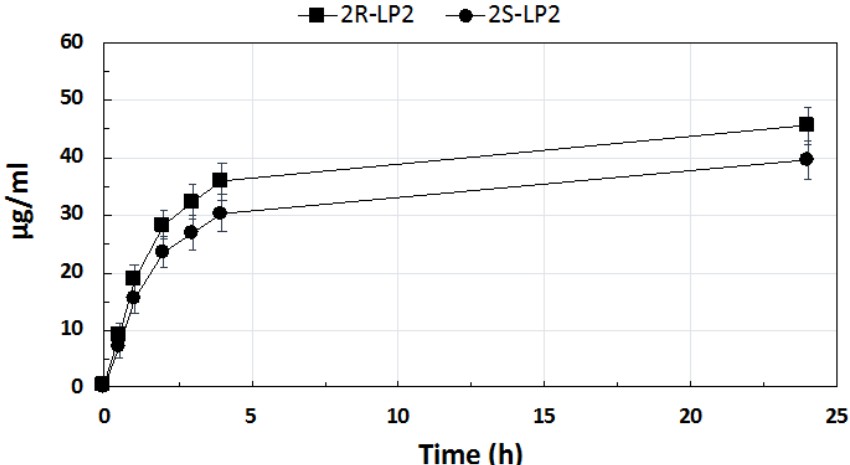

**Figure 4.** In vitro release from solid lipid nanoparticles of 2*R*-LP2 and 2*S*-LP2.

LP2 diastereoisomers' release was supposed to occur exclusively from the SLNs core owing to their lipophilicity, which prevented their solubilization in the aqueous medium [28].

After 24 h, the percentage of LP2 released, calculated as the mixture of 2R and 2S diastereoisomers, was 50.2%. Both diastereoisomers showed an initial fast release followed by a second phase of a slower release. Similar in vitro release profiles have already been reported by other authors for lipophilic drugs loaded into SLNs [29,30]. It is interesting to note that, although the release profile of both diastereoisomers was similar, the 2*R*-LP2 release rate was greater than that of 2*S*-LP2. As previously mentioned, diastereoisomers share identical molecular formulas, atom-to-atom linkages, and bonding distances, but the different spatial disposition of the chemical groups bonded to the chiral center could affect their ability to interact with the surrounding environment. Calculating Log *p* values of

2*R*- and 2*S*-LP2 using Marvin 20.11.0, ChemAxon (https://www.chemaxon.com, accessed on 24 March 2021), the same value (Log P 4.61) for both isomers was obtained. On the contrary, the lipophilic index, which stemmed from the interaction between the analyzed compound and the stationary (lipophilic) and mobile (hydrophilic) phases in the HPLC column, highlighted that 2*S*-LP2 was more lipophilic (Log K 0.66) than 2*R*-LP2 (Log K 0.60), thus suggesting a greater affinity of 2*S*-LP2 for the lipid core of SLNs, which, in turn, could lead to a slower release from the nanoparticles.

To point out the physico-chemical differences between 2*R*- and 2*S*-LP2, the HPLC method developed in this study aimed at finding the best conditions for the chromatographic separation of the two 2*R*/*S*-LP2 diastereoisomers on an achiral column, also obtaining a good peak shape and optimal retention, according to the recommendations by the International Conference on Harmonization (ICH) [25]. Reverse-phase chromatography (RPC) has become the predominant technique of analytical chromatography due to numerous advantages. The operative retention mechanisms operating in RPC permit excellent separations due to the possibility of varying pH and allow the use of organic modifiers and additives [31].

However, the analysis of basic compounds, such as LP2 diastereoisomers, poses difficulties in RPC analysis. In fact, preliminary analysis of LP-2 diastereoisomers in RPC provided large peaks with strong tailing that resulted in almost complete overlapping of the 2*R*/*S*-LP2 peaks.

In RPC, the problem of poor column efficiency and tailing peaks is attributed to mixed mechanisms involving the residual silanol groups (pKa 7–9) present in the C18 phase (40–50% in a typical C18-endcapped column). Hydrophobic interactions play a fundamental role, but a secondary mechanism involving ion exchange between analytes and the residual silanol groups can take place, as well. The slower sorption–desorption kinetics of silanol ion-exchange sites is responsible for tailing and poor efficiency (kinetic tailing) [32–34]. For these reasons, the effect of pH and the use of additives were evaluated as important factors to improve the separations and peak symmetry of 2*R*/*S*-LP2 diastereoisomers.

In order to suppress kinetic tailing due to the interaction between the basic nitrogen of LP2 and the residual silanol groups, mobile phases with a pH in the range of 2.5–3.5 were assessed. At this acidic pH, both the silanol groups and basic nitrogen are protonated, resulting in a weakening of their mutual interaction. To improve the peak symmetry, a tertiary amine, namely, 0.3% triethylamine (TEA), was used as an additive, as such bases have been proven to shield the silanol groups at an acidic pH [35,36], avoiding any residual interaction.

However, a major drawback of working at a low mobile-phase pH is that analytes exhibit low retention because basic molecules in the protonated form are more hydrophilic and have lower interaction with the lipophilic C18 stationary phase, with loss of selectivity, especially for very similar molecules, such as LP2 diastereoisomers [37].

For these reasons, the pH has been appropriately adjusted using trifluoroacetic acid, as this acid is a good ion-pairing reagent capable of increasing the retention of basic analytes, compensating for the loss of retention due to the low pH. Perfluorinated acids not only neutralize the positively charged nitrogen groups of the analytes by decreasing their hydrophilicity but are also able to improve the affinity for the hydrophobic stationary phase, thus increasing the retention.

To further improve the resolution of the 2*R*- and 2*S*-LP2 peaks, the behavior of the solvent eluents methanol (MeOH) and acetonitrile (MeCN) was also carefully evaluated. MeOH is mainly a proton-donor solvent with relatively high dipolar properties. MeCN possesses the largest dipole moment among the modifiers but is characterized by weak properties as a proton acceptor and much weaker properties as a proton donor.

It has been demonstrated that, in reverse phase analysis, analytes can interact with the stationary phase by means of solvation complexes; therefore, the solvent composition is crucial to obtain a good separation. Nigam et al. [38] reported that the hydrogen-bonding capacity for different solvent compositions of MeCN and MeOH was the following: methanol ≅ water > methanol–water complex >> acetonitrile–water complex ≅ acetonitrile.

Optimal separation of the two LP2 diastereoisomers peaks was obtained using methanol/TEA 0.3% (pH = 3 with TFA acid) as an eluent, while the use of acetonitrile significantly worsened the resolution between the two diastereoisomers. The 2-methoxy groups of LP2 diastereoisomers could be involved in proton–acceptor interactions with –OH groups of water and methanol of the eluent, but such interactions could not occur with acetonitrile, whose weaker proton-donor properties led to poor interactions with the methoxy group. The good resolution obtained with methanol as an eluent could be due to these solvation processes that could take place at different extents for the two LP2 diastereoisomers. The methoxy moiety of the 2*S*-LP2 diastereoisomer could be better solvated compared to the 2*R*-LP2 isomer; therefore, 2*S*-LP2 was relatively more hydrophobic and could interact more strongly with C18's stationary phase compared to 2*R*-LP2. This different behavior of 2*R*- and 2*S*-LP2 could be responsible for the complete resolution obtained using MeOH-based eluents.

The effect of column temperature in the range of 20–50 °C to obtain the best peak shape was also evaluated. As no significant change was recorded, analyses were performed at 22 °C.

In summary, the best conditions were obtained using a reversed-phase column, Knauer Eurosphere II 100–3 C18 (150 mm × 4.60 mm, 5.0 μm), using an isocratic mobile phase consisting of 50% MeOH and 50% TEA at 0.3% (pH = 3 with TFA acid) with a flow rate of 1.0 mL/min at 22 °C. Under these conditions, a baseline resolution of the two LP2 diastereoisomers was obtained with a value of R = 1.80 (USP, Ph. Eur.), with an acceptable peak shape (tailing factor = 1.18, calculated according to USP and Ph. Eur.). The retention times of 2*R*- and 2*S*-LP2 were approximately 9.0–9.5 and 10.0–10.5 min, respectively (Figure 5).

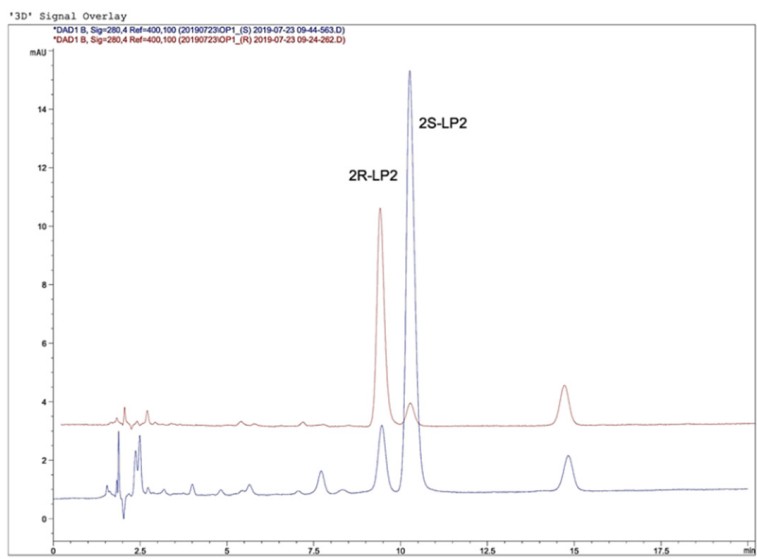

**Figure 5.** Chromatograms of 2*R*-LP2 and 2*S*-LP2.

Identification of the 2*R*/*S*-LP2 diastereoisomers was performed by comparing the retention times and UV spectra of the analyzed samples with those of standards. Furthermore, peak purity tests [39,40] were used to demonstrate the absence of coeluting peaks. Peak purity was determined with OpenLab software (Agilent Technologies) using photodiode-array detector spectra.

The calibration curve was constructed using seven calibrators for each diastereoisomer. The method was linear in the range of 60.00–0.54 μg/mL for 2*R*-LP2 and 44.00–0.56 μg/mL for 2*S*-LP2 (Table 2). Calibration curves were evaluated at the beginning of six consecutive days (*n* = 6). The equation of the calibration curve gave correlation coefficients higher than 0.999 for both diastereoisomers (Table 2).

**Table 2.** Calibration data, limit of detection (LOD), and limit of quantification (LOQ) obtained for HPLC analysis of 2*R*/*S*-LP2.

| Isomer | Regression Equation y = ax + b y = Peak Area, x = µg/mL | R² | Linear Range (µg/mL) | LOD (µg/mL) | LOQ (µg/mL) |
|---|---|---|---|---|---|
| 2R-LP2 | y = 4.13712028x −0.269608 | 0.99982 | 60.00–0.54 | 0.18 | 0.54 |
| 2S-LP2 | y = 4.12994575x −0.9244433 | 0.99956 | 44.00–0.56 | 0.18 | 0.56 |

The limit of detection (LOD) and limit of quantification (LOQ) were established with an S/N ratio of 3:1 and 10:1, respectively (*n* = 6). The LOD values were 0.18 µg/mL for both 2*R*/*S*-LP2 diastereoisomers, while the LOQ values were 0.54 µg/mL and 0.56 µg/mL for 2*R*- and 2*S*-LP2, respectively (Table 2).

The precision and accuracy were determined at three concentration levels in the dialysis medium consisting of water/ethanol (80/20 *v*/*v*) spiked with three known amounts within the calibration range by assaying replicates (*n* = 6) of 2*R*-LP2 (60.46, 8.75, and 0.54 µg/mL, Table 3) and 2*S*-LP2 (43.84, 8.75, and 0.54 µg/mL, Table 4), and processed as unknowns. To determine intra- and inter-day precision and accuracy, samples were analyzed in replicates on the same day and on three different days, respectively. Intra- and inter-day precision was reported as the relative standard deviation (RSD%), with an acceptability limit set at ≤15%. Accuracy was calculated by a comparison of mean assay results with the nominal concentrations, with acceptability limits set at ±15%. The intra-day and inter-day recoveries ranged from 96.30 to 104.57% for the two diastereoisomers, while the precision expressed as RSD% was less than 15% in both cases.

**Table 3.** Accuracy and precision of 2*R*-LP2 determination by HPLC. RSD = relative standard deviation. * *n* = 6.

| Spike Level (µg/mL) | Amount * (µg/mL) | Recovery (%) | RSD (%) |
|---|---|---|---|
| | | Intra-day | |
| 60.46 | 60.81 ± 0.63 | 100.58 | 1.04 |
| 8.75 | 8.91 ± 0.28 | 101.83 | 3.14 |
| 0.54 | 0.56 ± 0.07 | 103.70 | 12.50 |
| | | Intra-day | |
| 60.46 | 60.46 ± 0.97 | 102.73 | 1.56 |
| 8.75 | 9.15 ± 0.32 | 104.57 | 3.50 |
| 0.54 | 0.52 ± 0.07 | 96.30 | 13.46 |

**Table 4.** Accuracy and precision of 2*S*-LP2 determination by HPLC. RSD = relative standard deviation. * *n* = 6.

| Spike Level (µg/mL) | Amount * (µg/mL) | Recovery (%) | RSD (%) |
|---|---|---|---|
| | | Intra-day | |
| 43.84 | 43.84 ± 0.51 | 99.57 | 1.17 |
| 8.75 | 8.91 ± 0.13 | 101.83 | 1.46 |
| 0.54 | 0.53 ± 0.05 | 98.15 | 9.43 |
| | | Inter-day | |
| 43.84 | 43.77 ± 0.53 | 99.84 | 1.21 |
| 8.75 | 8.88 ± 0.15 | 101.49 | 1.69 |
| 0.54 | 0.55 ± 0.06 | 101.85 | 10.91 |

The development of an HPLC method like that described in this manuscript could be useful to assess the pharmacokinetic profile of other basic, synthetic, dual-target ligands such as piperazine derivatives designed to interact with serotonin receptors [41].

## 4. Conclusions

The dual-target MOPr/DOPr anti-nociceptive ligand LP2 shows an interesting pharmacological fingerprint, but its further development as a therapeutic agent for the management of acute and persistent pain is hindered by its high lipophilicity. Drug incorporation into SLNs has been proposed as a useful strategy to improve the unfavorable active ingredient features such as poor water solubility and stability. Therefore, in this work, LP2 was loaded into SLNs, and the technological properties of the resulting colloidal dispersions were assessed. As LP2 is a racemic mixture of two diastereoisomers (2$R$/2$S$-LP2), an HPLC method was developed to assess LP2 in vitro release from SLNs. Such a method allowed the successful separation of the 2$R$/$S$-LP2 diastereoisomers using a conventional C18 column with a mobile phase composed of MeOH/TEA 0.3% (pH = 3 with TFA). The incorporation of LP2 into SLNs provided colloidal dispersions with technological properties (small particle size, low polydispersity index, and good stability) that make them suitable for both parenteral and transdermal administration. Therefore, the strategy of loading LP2 into SLNs could be a useful tool to design formulations to perform further investigations on the pharmacological activity of this anti-nociceptive agent.

**Author Contributions:** Conceptualization, L.M.; methodology, L.M., A.S. and L.P.; validation, L.M., A.S. and L.P.; formal analysis, L.M., A.S., L.P., C.P., M.G.S. and R.T.; investigation, L.M.S., M.L. and M.G.S.; resources, L.P., A.S., M.G.S. and L.M.; data curation, L.M., A.S. and L.P.; writing—original draft preparation, L.M., L.P., A.S., M.G.S., C.P. and R.T; X.X.; writing—review and editing, L.M.; visualization, L.M.S., M.L. and M.G.S.; supervision, L.M.; project administration, L.M., A.S. and L.P.; funding acquisition, L.P., A.S. and L.P. All authors contributed equally to this work. All authors have read and agreed to the published version of the manuscript.

**Funding:** This research was funded by the University of Catania PdR2016-2018–UPB 57722172104.

**Institutional Review Board Statement:** Not applicable.

**Informed Consent Statement:** Not applicable.

**Data Availability Statement:** Data is contained within the article.

**Conflicts of Interest:** The authors declare no conflict of interest.

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
