# Peer review of "Solid Lipid Nanoparticles as Carriers for the Synthetic Opioid LP2: Characterization and In Vitro Release"

_applsci, doi:10.3390/app112110250_

Round 1
Reviewer 1 Report
Although the manuscript is well organized, the experimental part is sufficiently detailed and the results are clear, the main issue in is the absence of route of administration, making it difficult to understand the specific objective. The title, the abstract, the introduction, discussion and conclusions should clearly state the mode of administration intended for these solid lipid nanoparticles. The route of administration is a critical parameter in the development of drug delivery systems. I think these lipid carriers are designed for enhance transdermal delivery of the antinociceptive agent. this must be clearly stated in throughout the manuscript.
Adding the missing part will make the manuscript more interesting to readers including researchers, students and people in industry
Author Response
Although the manuscript is well organized, the experimental part is sufficiently detailed and the results are clear, the main issue in is the absence of route of administration, making it difficult to understand the specific objective. The title, the abstract, the introduction, discussion and conclusions should clearly state the mode of administration intended for these solid lipid nanoparticles. The route of administration is a critical parameter in the development of drug delivery systems. I think these lipid carriers are designed for enhance transdermal delivery of the antinociceptive agent. this must be clearly stated in throughout the manuscript.
Adding the missing part will make the manuscript more interesting to readers including researchers, students and people in industry
Answer
We would like to thank the reviewer for reviewing our manuscript and for the valuable comments.
We agree with the reviewer that the mode of administration intended for the solid lipid nanoparticles (SLNs) proposed in this study should be better specified. As pointed out in “Featured Application”, such SLNs were intended for parenteral administration. However, as the reviewer noticed, such solid lipid nanoparticles could be suitable for transdermal administration as well.
Therefore, we amended:
1) the abstract stating “In this work, to allow designing aqueous formulations of LP2 for parenteral administration, solid lipid nanoparticles (SLNs) were investigated as LP2 nanocarriers.”
2) the introduction stating “To address the issue of LP2 low water solubility allowing the design of formulations for parenteral administration, in the present work, this dual-target MOPr/DOPr anti-nociceptive ligand was incorporated into SLNs and the resulting colloidal dispersions were assessed to determine their technological properties (mean particle sizes, polydispersity index, zeta potential, morphology, stability) and in vitro LP2 release.”
3) the conclusions stating “The incorporation of LP2 into SLNs provided colloidal dispersions with technological properties (small particle size, low polydispersity index, good stability) that make them suitable for both parenteral and transdermal administration.
We did not specify both the potential administration routes in the title of the manuscript because we believe that this would make the title too long and less clear.

Reviewer 2 Report
The development of new therapeutic forms is a promising task, especially for hydrophobic compounds. The use of amphiphilic lipids is a good solution to this problem. The manuscript proposes a new formulation of the opioid peptide receptors, which based on solid lipid nanoparticles. They have good stability and biocompatibility. It would be good to discuss the stability of the proposed fomulation in the presence of amphiphilic albumin, which is found in high concentration in the blood, as well as its availability to cellular receptors. Nevertheless, the manuscript will be of interest to biotechnologists and pharmacists and may be accepted for publication.
Author Response
The development of new therapeutic forms is a promising task, especially for hydrophobic compounds. The use of amphiphilic lipids is a good solution to this problem. The manuscript proposes a new formulation of the opioid peptide receptors, which based on solid lipid nanoparticles. They have good stability and biocompatibility. It would be good to discuss the stability of the proposed fomulation in the presence of amphiphilic albumin, which is found in high concentration in the blood, as well as its availability to cellular receptors. Nevertheless, the manuscript will be of interest to biotechnologists and pharmacists and may be accepted for publication.
Answer
We would like to thank the reviewer for reviewing our manuscript and for the valuable comments. We agree with the reviewer that studies investigating the interactions between LP2 loaded solid lipid nanoparticles and albumin as well as LP2 availability at cellular level would support the usefulness of the strategy proposed in this work. As reported in the manuscript, LP2 is a diastereoisomeric mixture; therefore, we needed to validate a suitable analytical method to identify and to quantify each diasteroisomer prior to performing any investigation involving LP2 interactions with a biological environment. Therefore, in the present work, we focused on the technological characterization of the proposed nanocarriers and on the development and validation of an analytical method that could allow us performing further in vitro and in vivo studies involving LP2.

Reviewer 3 Report
In the current article the authors developed SLNs for the loading and delivery the synthetic anti-nociceptive ligand LP2. The study is well organised but it lacks in vivo data that could support the sued of SLNs which are a novel drug delivery system. Nevertheless the article is publishable after minor revision as the authors need to address the following:
- My major objection is that the composition of the nanoparticles is related to Nanostructured Lipid Carriers and not SLNs. The glyceryl oleate is the oil phase that the which is a typical feature for NLCs
- The particle size of empty and loaded SLNs doesn't show any difference which is obvious from the SDs of the measured particle (despite the PIT difference)
- Fig. 2 is of bad quality as the x-axis is impossible to read. The authors should include the integrated ΔΗ values of the DSC software. The enthalpy of load seems to be much greater compared to the empty. It is also unusual that the incorporation of the active will result in sharper endotherm.
- How the DT/2 was estimated?
- There must be an issue with the TEM images as the upper two particles in Fig. 3a are identical with those in Fig. 3b. Did the authors scanned the same area?
- Please add SDs in Fig. 4 for the release profiles and discuss the difference with other publications of SLNs (https://doi.org/10.1016/j.ijpharm.2015.07.042, https://doi.org/10.3109/02652048.2010.529948)
- For HPLC validation the regression equation should of the y = ax form and not the one presented in Table 2
Author Response
In the current article the authors developed SLNs for the loading and delivery the synthetic anti-nociceptive ligand LP2. The study is well organised but it lacks in vivo data that could support the sued of SLNs which are a novel drug delivery system. Nevertheless the article is publishable after minor revision as the authors need to address the following:
Answer
We would like to thank the reviewer for reviewing our manuscript and for the valuable comments. We agree with the reviewer that in vivo studies would support the usefulness of the nanocarriers investigated in this work. However, as the drug loaded into SLNs was a diastereoisomeric mixture, we needed a reliable analytical method to identify and to quantify each diasteroisomer prior to performing any pharmacokinetic study. Therefore, in this work we focused our attention on the characterization of the nanocarriers and on the development of a suitable and reliable analytical method.
- My major objection is that the composition of the nanoparticles is related to Nanostructured Lipid Carriers and not SLNs. The glyceryl oleate is the oil phase that the which is a typical feature for NLCs
Answer
In this work, we prepared lipid nanoparticles using cetyl palmitate as solid lipid and oleth-20 and glyceryl oleate as surfactant and co-surfactant, respectively. Glyceryl oleate is a co-emulsifier with low HLB value (HLB = 3) and we have already reported lipid nanoparticles obtained using glyceryl oleate as co-surfactant (DOI:10.1016/j.ijpharm.2012.05.046; DOI: 10.3109/03639045.2010.539231). As glyceryl oleate was used as co-surfactant and cetyl palmitate was a solid lipid, we defined these nanocarriers solid lipid nanoparticles. Due to the reviewer comment, we realized that the reader could misunderstand the use of glyceryl oleate, therefore we specified in the text (section 2.2. Preparation of solid lipid nanoparticles (SLNs) the use of glyceryl oleate as follows:
“SLNs were prepared using the phase inversion temperature (PIT) as previously reported [26,27]. The oil phase components were CP (7,0% w/w, solid lipid), oleth-20, (8,7% w/w, surfactant), GO (4,4 % w/w, co-surfactant) and LP2 (1,1 % w/w) while the aqueous phase consisted of saline solution (NaCl 0,90 % w/w).”
- The particle size of empty and loaded SLNs doesn't show any difference which is obvious from the SDs of the measured particle (despite the PIT difference)
Answer
We agree with the reviewer that empty and LP2 loaded SLNs had similar particle size as we reported in the text that “Unloaded SLNs, prepared by the PIT method, showed small mean particle size, which was not affected by LP2 loading”.
- 2 is of bad quality as the x-axis is impossible to read. The authors should include the integrated ΔΗ values of the DSC software. The enthalpy of load seems to be much greater compared to the empty. It is also unusual that the incorporation of the active will result in sharper endotherm.
Answer
We apologize for the inconvenience. We replaced figure 2 with a figure of better quality.
ΔΗ values obtained by the DSC software have already been reported in the text of the manuscript (ΔH was -11.90 J/g in unloaded SLNs and -18.02 J/g in LP2 loaded SLNs). As we analyzed 100 μl samples, it is evident that the integrated ΔΗ values of the DSC software were -1190 mJ for unloaded SLNs and -1802 mJ for LP2 loaded SLNs. However, due to the reviewer’s comment, we realized that there was a typo, as -12.90 should read -11.90. Therefore, we amended the text accordingly.
As the reviewer noticed, the enthalpy of LP2 loaded SLNs was greater compared to that of unloaded SLNs. As reported in the manuscript, we attributed this behavior to an increase of the lipid cooperativity due to the incorporation of LP2 into the SLNs.
- How the DT/2 was estimated?
Answer
Δ T ½ was calculated by the DSC software as temperature width at half peak height. We omitted this information but due to reviewer comment, we inserted in the text how Δ T ½ was calculated as follows:
ΔH was -11.90 J/g in unloaded SLNs and -18.02 J/g in LP2 loaded SLNs and a decrease of ΔT1/2 (calculated by the DSC software as temperature width at half peak height) from 5.99 °C in unloaded SLN to 4.40 °C in LP2 loaded SLN was observed.
- There must be an issue with the TEM images as the upper two particles in Fig. 3a are identical with those in Fig. 3b. Did the authors scanned the same area?
Answer
The TEM images reported in Fig. 3 a and b were obtained from different samples, as illustrated in the legend of Fig. 3. However, to avoid that the reader could share the same doubt, we changed the TEM images in Fig. 3 b.
- Please add SDs in Fig. 4 for the release profiles and discuss the difference with other publications of SLNs (https://doi.org/10.1016/j.ijpharm.2015.07.042, https://doi.org/10.3109/02652048.2010.529948)
Answer
As requested by the reviewer, we inserted SDs in Fig. 4.
The reviewer suggested discussing our results in comparison to other already published papers, citing the following manuscripts: 1) Delivery of retinoic acid to LNCap human prostate cancer cells using solid lipid nanoparticles by Akanda, Mushfiq H., Rai, Rajeev, Slipper, Ian J., Chowdhry, Babur Z., Lamprou, Dimitrios, Getti, Giulia, Douroumis, Dennis (DOI: 10.1016/j.ijpharm.2015.07.042); 2) Preparation and characterization of ibuprofen solid lipid nanoparticles with enhanced solubility by Sriharsha Gupta Potta, Sriharsha Minemi, Ravi Kumar Nukala, Chairmane Peinado, Dimitrios A. Lamprou, Andrew Urquhart and D. Douroumis (DOI: 10.3109/02652048.2010.529948). These manuscripts share two authors.
The first manuscript reported in vitro release experiments performed using a method similar to that described in our work (dialysis) but the second one investigated the dissolution rate of lyophilized SLN using a USP/Ph.Eur. paddle dissolution apparatus, which is a totally different method. Therefore, we believe that the second paper cited by the reviewer is not relevant and it cannot be taken into account to compare our results to literature data.
To comply with the reviewer’s request to discuss the results of our in vitro studies in comparison to literature data, we inserted in the text the following sentence:
“Similar in vitro release profiles have already been reported by other authors for lipophilic drugs loaded into SLNs [33,34]”.
We inserted the following references:
- Akanda, M.H.; Rai, R.; Slipper, I.J.; Chowdhry, B.Z.; Lamprou, D.; Getti, G.; Douroumis, D. Delivery of retinoic acid to LNCap human prostate cancer cells using solid lipid nanoparticles. Int J Pharm. 2015, 493, 161-71. DOI: 10.1016/j.ijpharm.2015.07.042.
- Venkateswarlu, V.; Manjunath, K. Preparation, characterization and in vitro release kinetics of clozapine solid lipid nanoparticles. J. Control Release, 2004, 95, 627-638. DOI: 10.1016/j.jconrel.2004.01.005.
We amended the references’ list accordingly.
- For HPLC validation the regression equation should of the y = ax form and not the one presented in Table 2.
Answer
We could not report the linear regression equation in the y = ax form (equation of a line passing through the origin), as suggested by the reviewer because the regression analysis gave us a linear equation in slope-intercept form (y = ax + b) where 'a' is the angular coefficient and 'b' is the y-intercept. The line obtained from this equation intersects the x-axis at a point different from zero. To express the regression equation in the y = ax form, we had to force the line to pass through the origin, thus altering the true experimental data as the term b is due to the existence of a detection limit in any HPLC determination. For a better comprehension, we have inserted in Table 2 the type of equation obtained.

Round 2
Reviewer 1 Report
Accepted